# Calibration and Validation of Antenna and Brightness Temperatures from Metop-C Advanced Microwave Sounding Unit-A (AMSU-A)

**Banghua Yan** [1,*]**, Junye Chen** [2]**, Cheng-Zhi Zou** [1]**, Khalil Ahmad** [2]**, Haifeng Qian** [3]**, Kevin Garrett** [1]**, Tong Zhu** [2]**, Dejiang Han** [4] **and Joseph Green** [5]

[1] Satellite Meteorology and Climatology Division, Center for Satellite Applications and Research, National Oceanic and Atmospheric Administration (NOAA), 5830 University Research Court, College Park, MD 20740, USA; cheng-zhi.zou@noaa.gov (C.-Z.Z.); kevin.garrett@noaa.gov (K.G.)

[2] Global Science Technologies, Greenbelt, MD 20770, USA; Junye.Chen@noaa.gov (J.C.); khalil.ahmad@noaa.gov (K.A.); tong.zhu@noaa.gov (T.Z.)

[3] Earth System Science Interdisciplinary Center, University of Maryland, 5825 University Research Ct Suite 4001, College Park, MD 20740, USA; haifeng.qian@noaa.gov

[4] Office of Satellite and Product Operations (OSPO), NOAA, 4231 Suitland Road Suitland, MD 20746, USA; dejiang.han@noaa.gov

[5] Office of Projects, Planning, and Analysis (OPPA), NOAA, 1335 East West Highway, Silver Spring, MD 20910, USA; Phil.Green@noaa.gov

\* Correspondence: banghua.Yan@noaa.gov; Tel.: +1-301-683-3602

**Abstract:** This study carries out the calibration and validation of Antenna Temperature Data Record (TDR) and Brightness Temperature Sensor Data Record (SDR) data from the last National Oceanic and Atmospheric Administration (NOAA) Advanced Microwave Sounding Unit-A (AMSU-A) flown on the Meteorological Operational satellite programme (MetOp)-C satellite. The calibration comprises the selection of optimal space view positions for the instrument and the determination of coefficients in calibration equations from the Raw Data Record (RDR) to TDR and SDR. The validation covers the analyses of the instrument noise equivalent differential temperature (NEDT) performance and the TDR and SDR data quality from the launch until 15 November 2019. In particular, the Metop-C data quality is assessed by comparing to radiative transfer model simulations and observations from Metop-A/B AMSU-A, respectively. The results demonstrate that the on-orbit instrument NEDTs have been stable since launch and continue to meet the specifications at most channels except for channel 3, whose NEDT exceeds the specification after April 2019. The quality of the Metop-C AMSU-A data for all channels except channel 3 have been reliable since launch. The quality at channel 3 is degraded due to the noise exceeding the specification. Compared to its TDR data, the Metop-C AMSU-A SDR data exhibit a reduced and more symmetric scan angle-dependent bias against radiative transfer model simulations, demonstrating the great performance of the TDR to SDR conversion coefficients. Additionally, the Metop-C AMSU-A data quality agrees well with Metop-A/B AMSU-A data, with an averaged difference in the order of 0.3 K, which is confirmed based on Simultaneous Nadir Overpass (SNO) inter-sensor comparisons between Metop-A/B/C AMSU-A instruments via either NOAA-18 or NOAA-19 AMSU-A as a transfer.

**Keywords:** Metop-C advanced microwave sounding unit-A; radiometry; calibration and validation; inter-sensor calibration among Metop-A to -C; simultaneous nadir overpass (SNO)

## 1. Introduction

The European Meteorological Operational satellite program C (Metop-C) satellite, which was launched into low Earth orbit on 6 November 2018, carries the last NOAA Advanced Microwave Sounding Unit-A (AMSU-A). On 15 November 2018, nine days after the launch of the Metop-C satellite, the first day AMSU-A science data were received. The AMSU-A provides temperature soundings from the Earth's near surface to an altitude of about 42 km through measurements of Raw Data Record (RDR) with 15 channels from 23.8 to 89 GHz. Table 1 lists the AMSU-A main channel characteristics, which include the channel frequency, bandwidth, and radiometric temperature sensitivity or Noise Equivalent Differential Temperature (NEDT) for each of the 15 channels. Intensive calibration activities for the AMSU-A Raw Data Record (RDR) to derive Earth antenna Temperature Data Record (TDR) data have been conducted in the NOAA Center for Satellite Applications and Research (STAR) [1–4]. Since April 2019, the TDR data have been distributed to the user community through both the NOAA Office of Satellite and Product Operations (OSPO) and Production Distribution Access (PDA) for near-real time applications and the NOAA Comprehensive Large Array-data Stewardship System (CLASS) for long-term data analysis and applications. Additionally, the conversion coefficients from TDR to Sensor Data Record (SDR) (brightness temperatures) data were derived in [5]. Today, Metop-C AMSU-A TDR and SDR data are successfully applied to a series of Environmental Data Record (EDR) retrieval systems and are also assimilated into the NOAA National Weather Service Global Forecast System (personal communication with Andrew Collard), the U.S. Navy Global Environmental Model (NAVGEM) (personal communication with Ruston Ben) and the European Centre for Medium-Range Weather Forecasts (ECMWF) global forecast system (personal communication with Niels Bormann).

**Table 1.** Advanced Microwave Sounding Unit-A (AMSU-A) instrument specifications [6].

| Channel Index | Center Frequency (MHz) | Central Frequency Stability (MHz) | Bandwidth (MHz) | Polarization | Measured 3-db Beamwidth [1,2] (°) | Temperature Sensitivity ($NE\Delta T$) (K) |
|---|---|---|---|---|---|---|
| 1 | 23,800 | ±10 | 270 | V | 3.48 | 0.3 |
| 2 | 31,400 | ±10 | 180 | V | 3.52 | 0.3 |
| 3 | 50,300 | ±10 | 180 | V | 3.64 | 0.4 |
| 4 | 52,800 | ±5 | 400 | V | 3.40 | 0.25 |
| 5 | 53,596 ± 115 [3] | ±5 | 170 | H | 3.60 | 0.25 |
| 6 | 54,400 | ±5 | 400 | H | 3.44 | 0.25 |
| 7 | 54,940 | ±5 | 400 | V | 3.44 | 0.25 |
| 8 | 55,500 | ±10 | 330 | H | 3.44 | 0.25 |
| 9 | $f_0$ = 57,290.344 | ±0.5 | 330 | H | 3.32 | 0.25 |
| 10 | $f_0$ ± 217 [3] | ±0.5 | 78 | H | 3.325 | 0.4 |
| 11 | $f_0$ ± 322.2 ± 48 [4] | ±1.2 | 36 | H | 3.32 | 0.4 |
| 12 | $f_0$ ± 322.2 ± 22 [4] | ±1.2 | 16 | H | 3.32 | 0.6 |
| 13 | $f_0$ ± 322.2 ± 10 [4] | ±0.5 | 8 | H | 3.32 | 0.8 |
| 14 | $f_0$ ± 322.2 ± 4.5 [4] | ±0.5 | 3 | H | 3.32 | 1.2 |
| 15 | 89,000 | ±130 | 1500 | V | 3.56 | 0.5 |

[1] Specifications of 3 db bandwidth are within 3.3° ± 10%; [2] 3-db bandwidth data correspond to beam position 15; [3] the channel has double bands; [4] the channel has four bands.

The data, either in TDR or SDR, from AMSU-A instruments onboard various legacy satellites from NOAA-15 to NOAA-19, and from Metop-A to Metop-B, play an important role in EDR retrieval systems [7–11], climate analysis [12,13], and Numerical Weather Prediction (NWP) models [14–17]. Metop-C AMSU-A data continue be used in those important fields. The sufficient calibration and validation of Metop-C AMSU-A data becomes very necessary to ensure the accuracy of the data. Another important parameter that could affect the performance of AMSU-A observations is the instrument noise [18,19], i.e., Noise Equivalent Differential Temperature ($NE\Delta T$ or NEDT), which represents the smallest temperature difference that an instrument can distinguish when looking at Earth scenes. This parameter also helps weight satellite data by channel in the observation error covariance matrix used by satellite EDR product retrieval systems [7], as well as by NWP data assimilation systems [15,17]. In climate studies, instrument noise affects the detection of long-term climate trends of Earth scene temperature data [20–22]. In addition, the AMSU-A instrument possesses four separate space view (SV) positions, resulting in the selection of an optimal cold space view position

among them, prior to measurements for operational use. Therefore, this work describes, in detail, the calibration and validation process for Metop-C AMSU-A from the RDR to SDR via TDR, including, but not limited to, the following analyses: optimal cold space position selection, cold space calibration correction, calibration coefficients, postlaunch instrument NEDT performance, TDR and SDR data quality validation. The lunar intrusion correction algorithm, which is another important portion of the calibration, is studied separately. The derivation process of the conversion coefficients from TDR to SDR is presented in detail in [5], although the conversion equation is briefly described in this study.

This paper is organized as follows. The next section provides a brief description of AMSU-A instruments along with an optimal cold space position selection based on Metop-C AMSU-A System In-Orbit Verification (SIOV) data. In Section 3, we establish the Metop-C AMSU-A calibration methodology, comprising the radiometric calibration equation from RDR to TDR, the conversion equation from TDR to SDR, and the determination of required coefficients and parameters in the equations. In Section 4, we analyze the instrument NEDT trend by using the current Integrated Calibration/Validation System (ICVS) developed at STAR [7] and compare it with those of legacy AMSU-A instruments. Meanwhile, a new NEDT estimation method is implemented for comparison. Regarding the data quality assessment, we conduct this analysis in Section 5. This is formed first by using the Joint Center for Satellite Data Assimilation (JCSDA) Community Radiative Transfer Model (CRTM) [23] to investigate the AMSU-A antenna and brightness temperatures bias features. Inter-sensor comparisons are further given of AMSU-A antenna temperatures between Metop-C and each of Metop-A and Metop-B AMSU-A instruments. These are performed by using each of the NOAA-18 and NOAA-19 AMSU-A instruments as a transfer based on the existing Simultaneous Nadir Overpass (SNO) method [24] and some proper quality schemes applicable for microwave satellite measurements at surface-sensitive channels [25,26]. The final section summarizes the overall Metop-C AMSU-A calibration and validation results.

## 2. AMSU-A Instrument Description and Optimal Cold Space Position Selection

The Metop-C AMSU-A instrument, which was built by Northrop Grumman, is composed of two modules, A1 and A2, with three antenna systems, A1-1, A1-2 and A2. The A1-1 system contains channels 6–7 and 9–15; A1-2 contains channels 3–5 and 8; and the A2 system contains channels 1 and 2. During each scan cycle, which lasts 8 s, the instrument samples 30 Earth scene cells (beam positions) within a satellite scan angle of 48.333° from nadir on each side of the sub-satellite path, each of which is separated by 3.33° in a stepped-scan fashion [27]. These scan patterns and geometric resolutions translate to a 48-km diameter cell at nadir and a 2343-km swath width from an 870-km nominal orbital altitude. In addition, the instrument measures the radiation from two calibration targets in every scan cycle, i.e., the cosmic background radiation or cold space that is viewed immediately after the Earth has been scanned, and the internal blackbody calibration target or warm load that is viewed immediately after the cold space. As a result, every scan cycle contains three consecutive views: 30 Earth scenes, cold space and blackbody warm calibration measurements (see Figure 1).

As illustrated in Figure 1, the AMSU-A instrument possesses four separate space view (SV) positions, i.e., 83.3° (SV1), 81.67° (SV2), 80.0° (SV3), and 76.67° (SV4). In practice, however, an optimal cold space view position among them needs to be determined prior to measurements for operational use. This optimal SV position is assumed to produce cold counts with minimum contamination radiating from the spacecraft and Earth's limb, thus mostly providing a minimum averaged cold count per SV period. The optimal SV position for Metop-C AMSU-A is selected during the instrument SIOV early on-orbit verification (OV) period. This period covers 08:15am `Coordinated Universal Time (UTC)` on 19 November, 2018 to 14:21 UTC on 30 November, 2018, and is made up of observations of approximately 30 consecutive orbits (2 days) for each of the four positions (SV1 to SV4), with the exception of position 4, which has three consecutive days of measurements. In addition, two scanning modes were set up for the SV1 position, so the second was defined as 'SV1n' to distinguish it from the first. In total, five sets of data were collected (refer to Table 2). In addition, a few types of

signals unrelated to the change in SV position need to be removed in the measured cold count data sets during the above OV period, e.g., lunar contamination events, count outliers, variations due to instrument noise, diurnal and orbital variations due to instrument temperature change, and trends due to instrument warm-up. The data sets were pre-processed to catch features of cold counts, primarily due to the change in SV position.

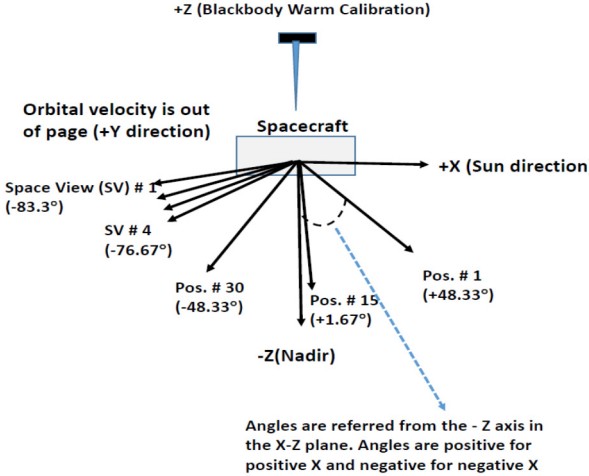

**Figure 1.** AMSU-A geometry sketch of a coordinator placed at the center of the instrument antenna [28].

**Table 2.** Daily averaged space view counts corresponding to each space view (SV) position and optimal selection for each channel of Metop-C AMSU-A instrument.

| Channel | SV1 * | SV2 * | SV4 * | SV 3 * | SV1n * | Optimal SV |
|---|---|---|---|---|---|---|
| 1 | 11,862.49 | 11,862.90 | 11,865.61 | 11,863.34 | 11,862.85 | SV1 |
| 2 | 11,351.19 | 11,351.23 | 11,353.70 | 11,350.86 | 11,351.80 | SV3 |
| 3 | 11,794.02 | 11,791.19 | 11,794.24 | 11,789.59 | 11,785.46 | SV1n |
| 4 | 12,694.09 | 12,694.11 | 12,696.86 | 12,695.06 | 12,692.76 | SV1n |
| 5 | 13,123.09 | 13,123.91 | 13,125.99 | 13,125.03 | 13,125.80 | SV1 |
| 6 | 12,335.72 | 12,335.36 | 12,336.37 | 12,337.13 | 12,337.19 | SV2 |
| 7 | 12,813.11 | 12,807.50 | 12,809.56 | 12,810.05 | 12,813.40 | SV2 |
| 8 | 12,196.05 | 12,196.08 | 12,196.09 | 12,200.60 | 12,199.81 | SV1 |
| 9 | 12,284.01 | 12,281.09 | 12,278.17 | 12,278.76 | 12,280.33 | SV4 |
| 10 | 12,184.44 | 12,183.84 | 12,183.69 | 12,186.46 | 12,189.38 | SV4 |
| 11 | 13,059.05 | 13,061.89 | 13,060.14 | 13,062.13 | 13,066.32 | SV1 |
| 12 | 12,820.44 | 12,821.88 | 12,819.51 | 12,819.92 | 12,823.72 | SV4 |
| 13 | 13,287.03 | 13,288.95 | 13,286.26 | 13,285.88 | 13,288.63 | SV3 |
| 14 | 12,760.81 | 12,764.91 | 12,764.76 | 12,766.90 | 12,771.17 | SV1 |
| 15 | 13,843.62 | 13,843.38 | 13,842.72 | 13,848.32 | 13,850.08 | SV4 |

* On-orbit verification dates corresponding to five SV data sets: 19 November 2018 to 21 November for SV1; 21 November 2018 to 23 November for SV2; 23 November 2018 to 26 November for SV4; 26 November 2018 to 28 Nov for SV3; 28 November 2018 to 30 November for SV1n (second cycle for SV1).

For demonstration, Figure 2 displays a time series of the data sets at channel 8, from original cold count measurements to the 'cold counts' after the corrections of the abovementioned signals, step by step, i.e., (a) original cold count measurements, (b) cold counts after lunar contamination removal (i.e., Lu-Rm), (c) 'cold counts' after removing count outliers from (b) (i.e., Lu-Ot-Rm), (d) 'cold counts' after filtering high frequent noise components from (c) (i.e., Lu-Ot-Hf-Rm), (e) 'cold counts' after mitigating diurnal and orbital variations due to instrument temperature change from (d) (i.e., Lu-Ot-Hf-Cy-Rm), and (f) 'cold counts' after removing trend due to instrument warm-up from (e) (i.e., Lu-Ot-Hf-Cy-Trd-Rm). Note that the impact of the lunar intrusion on the overall

cold counts is small during this period. The maximum magnitude of the lunar contamination is about 10 counts, which occurred on 27 November 2018. A similar procedure is applied to other channels. Therefore, for a given channel, the resulting data after the corrections are averaged for each SV position to produce the mean count per SV, as given in Table 2.

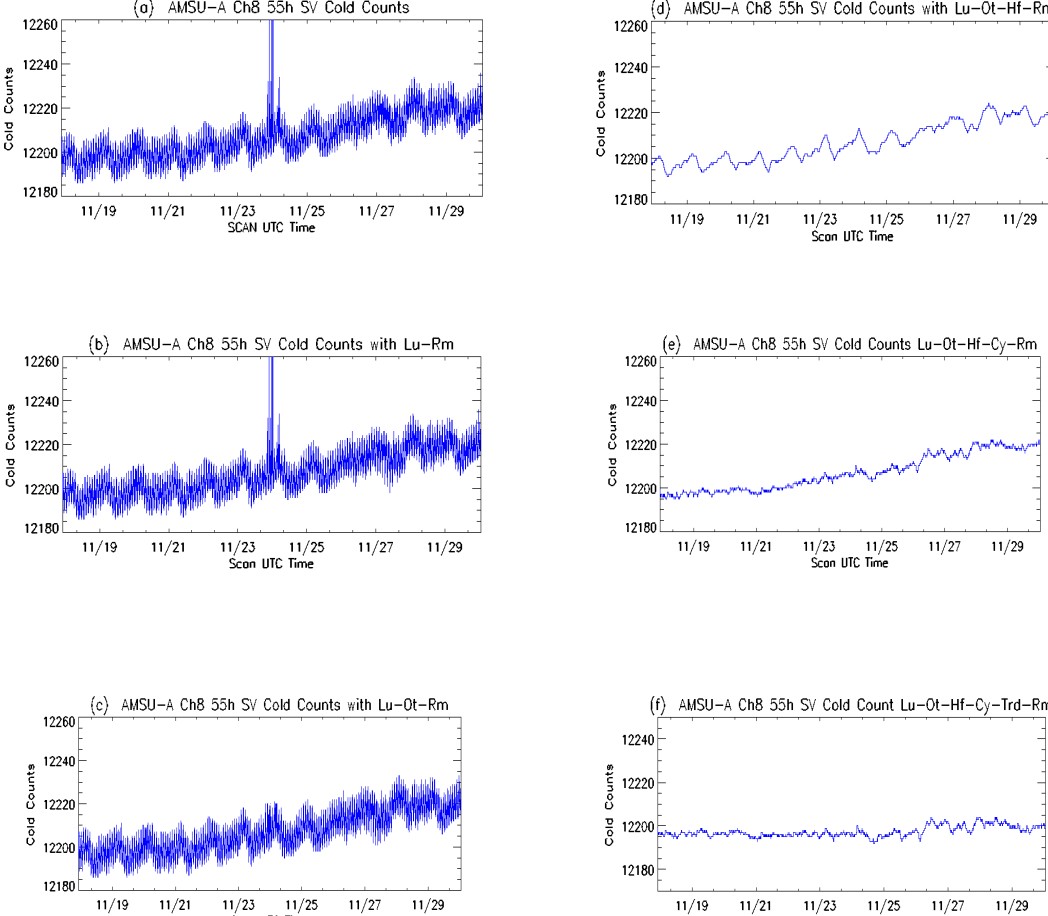

**Figure 2.** Cold count time series during November 19 0815 UTC and November 30 1421 UTC, 2018 for Metop-C AMSU-A channel 8 after a series of corrections of signals, as mentioned in the main text. (**a**) Original cold count measurements. (**b**) Cold counts after lunar contamination removal (i.e., Lu-Rm). (**c**) 'Cold counts' after removing count outliers from (**b**) (i.e., Lu-Ot-Rm). (**d**) 'Cold counts' after filtering high frequent noise components from (**c**) (i.e., Lu-Ot-Hf-Rm). (**e**) 'Cold counts' after mitigating diurnal and orbital variations due to instrument temperature change (i.e., Lu-Ot-Hf-Cy-Rm). (**f**) 'Cold counts' after removing trend due to instrument warm-up from (**e**) (i.e., Lu-Ot-Hf-Cy -Trd-Rm).

As shown in Table 2, for AMSU A1 channels, the frequency of the occurrence with the lowest averaged count is six times for SV1(n), twice for SV2, once for SV3 and four times for SV4. For AMSU-A2 channels, the averaged counts are very similar for all SV positions, although the counts for SV3 are slightly lower. As a result, the optimal cold position is SV1 (which is nearest to the satellite platform) for AMSU-A1 and SV3 for AMSU-A2. Since 30 November 1421 UTC, AMSU-A1 and A2 cold space positions have been switched to positions 1 and 3, respectively, to start regular measurements, which were conducted by the Metop-C flight team of the National Aeronautics and Space Administration (NASA) Goddard Space Flight Center (GSFC). Note that the choice for AMSU-A1 is the same as that of legacy AMSU-A instruments onboard NOAA-16, 18, Metop-A and –B, but the choice for AMSU-A2 is different from that of legacy AMSU-A2 instruments. The data used in the following analyses correspond to the selected optimal SV positions unless otherwise noted.

### 3. AMSU-A Calibration Methodology Description

The calibration methodology consists of a radiometric equation from RDR to TDR and a conversion equation from TDR to SDR. Basic equations are the same as previous studies for legacy AMSU-A instruments flown onboard NOAA-15–19, Metop-A and –B [29–32], except for the different calibration coefficients, nonlinearity and cold target calibration corrections, so the equations are only briefly described below. The equations are relevant to the channel frequency ($v$) and beam position (satellite zenith angle $\beta$), but those indices are typically omitted in this study for clarity unless otherwise noted.

#### 3.1. Calibration Equations

Two calibration measurements, i.e., cold space and warm load, are used to determine antenna temperatures via a radiometric calibration equation, as illustrated in the calibration scheme in Figure 3. In particular, the radiometric calibration equation converts the measured digitized radiometric scene counts $C_S$ (i.e., scene counts) to radiance $R_S$ for the Earth scene target using the following equation [33,34].

$$R_S = R_W + (R_W - R_C)\frac{\left(C_S - \overline{C}_W\right)}{\left(\overline{C}_w - \overline{C}_c\right)} + Q = R_{SL} + Q \tag{1}$$

with

$$R_{SL} = R_W + (R_W - R_C)\frac{\left(C_S - \overline{C}_W\right)}{\left(\overline{C}_w - \overline{C}_c\right)} = R_W + \frac{\left(C_S - \overline{C}_W\right)}{G} \tag{2}$$

$$Q = \mu(R_W - R_C)^2\frac{\left(C_S - \overline{C}_W\right)\left(C_S - \overline{C}_C\right)}{\left(\overline{C}_W - \overline{C}_C\right)^2} \tag{3}$$

$$G = \frac{\left(\overline{C}_W - \overline{C}_C\right)}{(R_W - R_C)} \tag{4}$$

where $R_S$ represents the radiometric scene radiance of individual channels, accounting for the nonlinear contribution due to an imperfect square law detector (see the line $\overline{CSW}$ in Figure 3); $R_{SL}$ denotes a linear two-point calibration equation with the assumption of a perfect detector (see the dash line $\overline{CS_LW}$ in Figure 3); $C_S$ is the radiometric count from the Earth scene target; $G$ is the channel calibration gain; $\overline{C}_W$ and $\overline{C}_C$ denote the averaged blackbody and space counts, respectively, over several calibration cycles (Appendix A). In addition, $R_W$ and $R_C$ denote the radiance corresponding to $T_W$ and $T_c$, respectively. $T_W$ denotes the platinum resistance thermometer (PRT) temperature of the warm load converted from measured radiometric counts, and its calculation and calibration are given in [1] and is also referred to Appendix B. The conversion coefficients from counts to PRT temperature are included in TDR data. $T_c$ is the cosmic temperature after certain correction, and $Q$ is the nonlinearity of the instrument's square law detector, which is a function of nonlinearity parameter μ (see discussions in Section 3.2). The variables, i.e., $R_C$, $R_W$, and $R_S$, in the above equations denote the radiance, represented in mW/(m²·sr·cm). In reality, by following the processing procedure for legacy AMSU-A measurements flown on the NOAA-15, -16, -17, and -18, -19, Metop-A and -B satellite platforms, the final output from TDR data for Metop-C are presented as the temperature instead of the radiance. Therefore, in the operational processing of Metop-C AMSU-A TDR data, a conversion between temperature and radiance is needed and is achieved by using the inverse of the Planck function in [35].

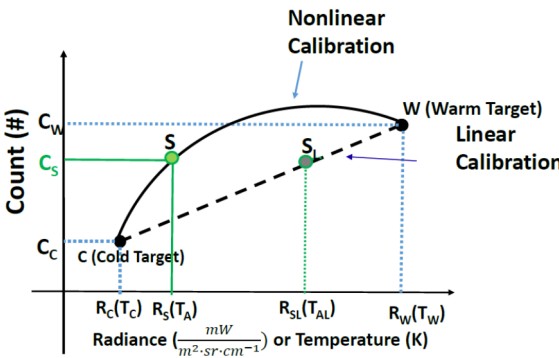

**Figure 3.** AMSU-A instrument linear and nonlinear calibration scheme.

The AMSU-A instrument is composed of two units (A1 and A2), and has three antenna systems, A1-1, A1-2 and A2, where the A1-1 system contains channels 6–7 and 9–15, the A1-2 contains channels 3–5 and 8 and the A2 system contains channels 1 and 2. Each of these systems consists of an offset parabolic reflector housed in a cylindrical shroud [27]. The temperature corresponding to $R_S$ is actually the antenna temperature ($T_A$), which is provided in the TDR data. $T_A$ usually contains antenna sidelobe contributions [5,28], antenna emissions and other radiation perturbations [5,34]. Hence, the brightness temperature of Earth scene $T_B$ needs be obtained from the antenna temperature after removing antenna sidelobe contributions, antenna emissions and other radiation perturbations, which are usually collectively defined as the antenna pattern correction for AMSU-A [28].

To understand the conversion from antenna temperature to brightness temperature, Equation (1) is expressed in temperature, i.e.,

$$T_A = T_W + (T_W - T_C)\frac{\left(C_S - \overline{C}_W\right)}{\left(\overline{C}_w - \overline{C}_c\right)} + Q_T, \tag{5}$$

where $Q_T$ is the nonlinearity of the instrument square law detector in temperature, converted from $Q$ in Equation (1).

The antenna pattern corrections and the recovery of brightness temperatures from measured antenna temperatures obtained from legacy AMSU-A radiometers were studied previously [28]. By taking advantage of existing algorithms [28,34], we have established a similar conversion from Earth scene antenna temperature in TDR to Earth scene brightness temperature in SDR for Metop-C AMSU-A [5]. According to [5], brightness temperatures are computed from antenna temperature using the following expression.

$$T_B(\beta) = \alpha_0(\beta)T_A(\beta) - \alpha_1(\beta) \tag{6}$$

with

$$\alpha_0(\beta) = 1.0 + \frac{f_C(\beta)}{f_E(\beta)} + \frac{\sigma f_{SAT}(\beta)}{f_E(\beta)} \tag{7}$$

$$\alpha_1(\beta) = \frac{f_C(\beta)T_C + \sigma f_{SAT}(\beta)T_{SAT}}{f_E(\beta)} \tag{8}$$

In these equations, $f_E(\beta)$, $f_C(\beta)$ and $f_{SAT}(\beta)$ represent the antenna pattern efficiencies over regions of Earth (main and sidelobes), cold space (sidelobes) and satellite spacecraft (sidelobes), respectively. The efficiency values are provided upon request from the authors. The satellite zenith angle $\beta$ (see Figure 1) is included to highlight that the three efficiencies are not constant with the beam position. The $\sigma$ is a scale factor to take into account the approximation of the near-field effect of the satellite platform in the antenna pattern correction, varying from 0.01 at channel 1 to 0.11 at channel 15, depending on the channel [28,36].

### 3.2. Determinations of Cold Space Calibration Correction and Nonlinearity

Three important variables, i.e., $T_c$, $Q$ and $\mu$, which are used in Equation (1), are determined in prelaunch, as introduced below.

$T_C$ represents the cold space brightness temperature, and is the cosmic temperature ($T_{Cosmic}$) after removing the correction of the antenna side lobe interference on cold space temperature via the Earth limb and spacecraft, as well the nonlinearity of the instrument square law detector. It is estimated by adding two correction terms to $T_{Cosmic}$ [6,37]:

$$T_C = T_{Cosmic} + \Delta T_C^{RJ} + \Delta T_C^{ER} \tag{9}$$

where $T_{Cosmic}$ is 2.72 K with an uncertainty of ±0.02 K. $\Delta T_C^{RJ}$, representing a correction using Planck's Radiation Law for the error introduced by the Rayleigh–Jeans (RJ) approximation and is given in the second column of Table 3 according to the analysis in [6,37].

**Table 3.** Bias correction for the cosmic cold background. In the table, $\Delta T_C^{ER}$ is computed using (10) corresponding to the selected optimal SV position.

| Channel Index | $\Delta T_C^{RJ}$ (K) | $\Delta T_C^{ER}$ (K) |
|:---:|:---:|:---:|
| 1 | 0.040 | 1.162 |
| 2 | 0.069 | 1.107 |
| 3 | 0.176 | 1.994 |
| 4 | 0.194 | 2.269 |
| 5 | 0.200 | 2.089 |
| 6 | 0.206 | 1.253 |
| 7 | 0.210 | 1.615 |
| 8 | 0.214 | 1.903 |
| 9–14 | 0.228 | 1.138 |
| 15 | 0.537 | 0.754 |

$\Delta T_C^{ER}$ in (9) signifies the contribution from the antenna side lobe interference with the Earth limb and spacecraft. By following the methodology in [5], $\Delta T_C^{ER}$ is computed using the following equation

$$\Delta T_C^{ER} = \left(1 - \epsilon_{Ref}\right)\left\{\frac{1}{N_\sigma}\left(f_E^{SV_i}\overline{T}_{ELB} + f_C^{SV_i}T_{Cosmic} + f_{SAT}^{SV_i}\sigma\overline{T}_{SAT}\right) - T_{Cosmic}\right\} + \epsilon_{Ref}T_{SAT}\,, \tag{10}$$

Approximately,

$$\Delta T_C^{ER} = \frac{\left(1 - \epsilon_{Ref}\right)}{N_\sigma}\left(f_E^{SV_i}\overline{T}_E + f_{SAT}^{SV_i}\sigma\overline{T}_{SAT}\right) + \epsilon_{Ref}\overline{T}_{SAT} \tag{11}$$

where $\epsilon_{Ref}$ denotes the emissivity of the AMSU-A reflector; $\overline{T}_{ELB}$ (=210 K) denotes an averaged Earth limb brightness temperature; $\overline{T}_{SAT}$ denotes an averaged instrument temperature. The quantity $N_\sigma = f_E(\beta) + f_C(\beta) + f_{SAT}(\beta)\sigma$ normalizes the contribution of energy by each radiation component. $f_E^{SV_i}$, $f_C^{SV_i}$, and $f_{SAT}^{SV_i}$ are the antenna efficiencies over the region of cold space, where the subscript '$i$' in $SV_i$ corresponds to a specific SV position with a defined beam position where i = 1~4. The values for 15 channels are provided in Table 4, which are computed using Metop-C AMSU-A pattern function data. The estimated $\Delta T_C^{ER}$ at all channels using (10) is shown in the third column of Table 3, where $\overline{T}_{SAT} = 300\tilde{K}$; $\overline{T}_E = 210$ K; $\epsilon_{Ref} \approx 0.0002$, 0.0006, 0.0004, 0.0004, 0.0005, 0.0003, 0.0004, 0.0006, 0.0003 and 0.0005 for the ten channels in the table according to the Northrop Grumman Electronic Systems (NGES) Calibration Log Book [6,37].

**Table 4.** Metop-C AMSU-A Antenna efficiencies (%) at four cold space view (SV) positions for 15 channels over regions of cold space, Earth, and satellite spacecraft. The selected SV positions are highlighted in the table.

| Ch. | FC (%) | | | | FE (%) | | | | FSAT (%) | | | |
|-----|------|------|------|------|------|------|------|------|------|------|------|------|
| | SV1 | SV2 | SV3 | SV4 | SV1 | SV2 | SV3 | SV4 | SV1 | SV2 | SV3 | SV4 |
| 1 | 99.06 | 99.00 | 99.08 | 99.09 | 0.49 | 0.54 | 0.5 | 0.53 | 0.52 | 0.49 | 0.48 | 0.44 |
| 2 | 99.23 | 99.17 | 99.25 | 99.26 | 0.36 | 0.43 | 0.39 | 0.41 | 0.42 | 0.39 | 0.37 | 0.34 |
| 3 | 98.45 | 98.31 | 98.45 | 98.45 | 0.85 | 1.00 | 0.88 | 0.90 | 0.70 | 0.68 | 0.67 | 0.64 |
| 4 | 98.19 | 98.04 | 98.21 | 98.22 | 0.97 | 1.14 | 1.00 | 1.02 | 0.84 | 0.81 | 0.80 | 0.77 |
| 5 | 98.52 | 98.40 | 98.54 | 98.55 | 0.87 | 1.01 | 0.90 | 0.92 | 0.63 | 0.60 | 0.58 | 0.55 |
| 6 | 98.99 | 98.93 | 99.00 | 99.00 | 0.53 | 0.61 | 0.55 | 0.57 | 0.49 | 0.47 | 0.46 | 0.44 |
| 7 | 98.67 | 98.66 | 98.70 | 98.72 | 0.68 | 0.71 | 0.71 | 0.73 | 0.66 | 0.63 | 0.61 | 0.57 |
| 8 | 98.76 | 98.60 | 98.76 | 98.76 | 0.79 | 0.95 | 0.82 | 0.83 | 0.48 | 0.46 | 0.44 | 0.42 |
| 9–14 | 99.07 | 99.02 | 99.09 | 99.09 | 0.48 | 0.54 | 0.50 | 0.52 | 0.46 | 0.45 | 0.43 | 0.41 |
| 15 | 99.52 | 99.49 | 99.52 | 99.52 | 0.27 | 0.27 | 0.27 | 0.27 | 0.27 | 0.26 | 0.25 | 0.25 |

Note that some uncertainties remain with the estimation of $\Delta T_C^{ER}$. Particularly, the calculation of $\Delta T_C^{ER}$ relies on an averaged Earth scene brightness temperature ($\overline{T}_{ELB}$) per channel. However, the Earth scene temperature can vary by location over the Earth. For example, the brightness temperature at channel 1 varies primarily between 180 K and 310 K, which could cause an error of up to one quarter of the original estimation. Theoretically, the $\Delta T_C^{ER}$ should be computed using the actual Earth scene temperature per location, thus producing a changeable correction along with the location. However, in the current operational processing system for all AMSU-A TDR observations, a fixed correction is used to reduce the contamination from the antenna sidelobe interference with the Earth limb and spacecraft.

Regarding the quantity $Q$, this represents the nonlinear contribution to $R_S$ due to an imperfect square law detector being a function of parameter μ. To quantify the magnitude of the instrument linearity performance, the $Q$ was estimated using prelaunch Metop-C AMSU-A Thermal Vacuum Chamber (TVAC) data sets. The TVAC data were taken at three instrument temperatures (see Table 5) and the scene target was cycled at each instrument temperature through six temperatures 84, 130, 180, 230, 280, and 330 K, respectively [6,37]. According to the analysis in [1], the maximum (absolute) Q values for Metop-C AMSU-A instruments are about 0.6 K. Consequently, Metop-C AMSU-A instrument nonlinearities at all channels exceed the specification since the specification requires Q = 0.5 K for channels 1, 2, and 15, and Q = 0.375 K for other channels. This indicates the significance of applying the nonlinearity correction in the instrument calibration process. Table 5 shows the values of parameter μ at three instrument temperatures (low, nominal, and high). After launch, the μ values at the actual on-orbit instrument temperatures are interpolated from these three values. For channels 9–14 (AMSU-A1-1), two sets of the μ parameters are provided; one set is for the primary Phase Locked-Loop Oscillators (PLLO) #1 and the other one for the redundant PLLO #2 phase.

Therefore, the Metop-C AMSU-A Earth scene antenna and brightness temperatures can be determined using Equation (1) and Equation (6), along with the prelaunch-determined coefficients and corrections that are provided in Tables 4 and 5. The following two sections focus on the assessment of the instrument noise performance and the derived TDR and SDR data quality, correspondingly, since launch to 15 November 2019.

**Table 5.** Nonlinearity parameters µ in dimension of $(m^2\text{-sr-cm}^{-1})/mW$ for 15 Metop-C AMSU-A channels, which were derived in [1].

| Ch. # | 1st Instrument Temperature | | 2nd Instrument Temperature | | 3rd Instrument Temperature | |
|---|---|---|---|---|---|---|
| 1 | 5.802 | | 5.600 | | 5.769 | |
| 2 | 2.236 | | 2.192 | | 2.145 | |
| 3 | 0.096 | | 0.100 | | −0.076 | |
| 4 | 0.881 | | 1.005 | | 0.969 | |
| 5 | 0.597 | | 0.724 | | 0.597 | |
| 6 | 3.309 | | 2.849 | | 2.146 | |
| 7 | 3.180 | | 2.698 | | 2.011 | |
| 8 | 0.574 | | 0.670 | | 0.569 | |
| | PLLO#1 | PLLO#2 | PLLO#1 | PLLO#2 | PLLO#1 | PLLO#2 |
| 9 | 3.011 | 2.988 | 2.598 | 2.594 | 2.02 | 2.248 |
| 10 | 3.391 | 3.298 | 2.915 | 2.927 | 2.27 | 2.517 |
| 11 | 3.031 | 3.047 | 2.748 | 2.801 | 2.225 | 2.461 |
| 12 | 3.115 | 3.184 | 2.915 | 2.942 | 2.426 | 2.659 |
| 13 | 3.106 | 3.107 | 2.817 | 2.944 | 2.43 | 2.66 |
| 14 | 3.075 | 3.157 | 3.007 | 3.035 | 2.4 | 2.773 |
| 15 | 1.216 | | 0.990 | | 0.710 | |

Notes: For AMSU-A1, the three instrument temperatures are −2, 18, and 38 °C; for AMSU-A2, the three instrument temperatures are −7, 11.5, and 30 °C.

## 4. Instrument Noise Performance Assessment

Currently, the on-orbit NEDT performance of AMSU-A and other microwave instruments is characterized typically using gain-based statistical methods. In gain-based methods, the NEDT is defined as the quotient of the fluctuation (standard deviation or overlapping Allan deviation) of warm counts and the calibration gain during one orbit of observations [18,38]. The gain denotes the averaged sensitivity of calibration counts per Kelvin [32] (also refer to Appendix C). In particular, the overlapping Allan deviation [18,34,39,40] is employed in the NOAA Integrated Calibration/Validation System (ICVS), which is briefly described in Appendix B. Recently, the gain-based methods have been revealed to over-estimate the instrument noise because of an overrated temperature sensitivity to warm counts due to the use of the gain [19]. Nevertheless, the ICVS gain method (namely the ICVS method) has been widely applied to all AMSU-A and Microwave Humidity Sounder (MHS) instruments onboard NOAA-15, -16, -17, and -18, -19, Metop-A, and -B AMSU-A. To comply with legacy AMSU-A instrument noise analysis, in this study, the ICVS method continues to be applied to Metop-C AMSU-A for the one-year noise performance assessment, albeit the new method is used for comparison. Several important conclusions are discovered from our results, as described below.

Firstly, the Metop-C AMSU-A instrument has a stable noise performance for all channels except for channel 3. For demonstration, Figure 4 displays the AMSU-A specification, prelaunch and on-orbit NEDT at 15 channels on the first day (5 November 2018), the 90th day (15 February 2019), and one year (15 November 2019) after the launch. The AMSU-A channel noises from 1 to 2 and 4 to 15 are within the specification and are also lower than or comparable to the prelaunch values. However, the channel 3 NEDT is unstable and gradually exceeds the specification. To better understand this feature, Figure 5a displays the time series of the channel 3 NEDT from 15 November 2018 to 15 November 2019. The NEDT was mostly within the specification (0.4 K) prior to 7 April 2019, but it rises to the order of 1 K, which exceeds the specification from this point onwards. This feature is attributed to noisy calibration target counts. Figure 5b,c display the time series of daily mean and standard deviation for the same time period for warm load counts and cold counts, respectively. The warm count standard deviation apparently rises with time after March 2019, which directly causes a high overlapping Allan deviation. Meanwhile, the cold counts increase more rapidly than the warm counts, thus producing a degraded gain with time. For example, as of 15 November 2019, the daily mean warm count, cold count and gain at channel 3 have been changed by approximately 22.7%, 40.3% and −38.2% (decrease), respectively, compared with the first day of the data (i.e., 15 November 2018). Therefore, the increased overlapping Allan deviation, but decreased gain, produces a high NEDT, as shown in Figure 5a.

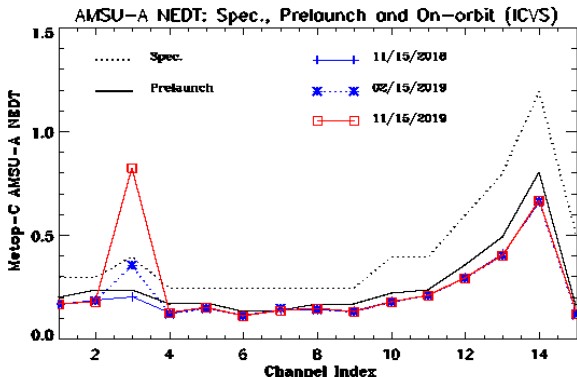

**Figure 4.** Metop-C AMSU-A specification, prelaunch and on-orbit noise equivalent differential temperature (NEDT) at 15 channels on the first day (15 November 2018), 90th day (15 February 2019), and one year (15 November 2019) after the launch.

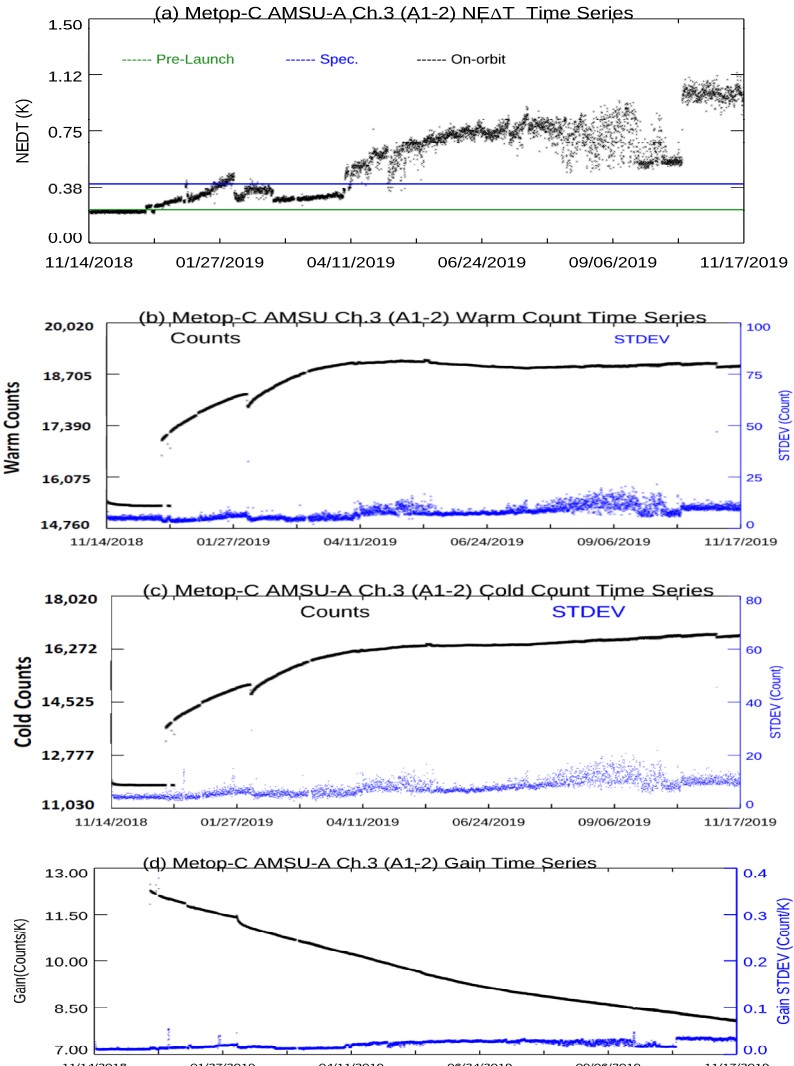

**Figure 5.** Time series of on-orbit NEDT, warm counts, cold counts and calibration gain for Metop-C AMSU-A channel 3 from 15 November 2018 to 15 November 2019, where the standard deviation of daily variations per parameter is included as the second *Y*-axis in the figures. (**a**) On-orbit NEDT. (**b**) Warm counts. (**c**) Cold counts. (**d**) Calibration gain.

Secondly, the Metop-C instrument exhibits slightly smaller noise values than two legacy AMSU-A instruments onboard Metop-A and -B satellites with some exceptions at channel 3. For demonstration, Figure 6a shows the results on 15 November 2019 among Metop-A to -C AMSU-A instruments, where Metop-A channels 7 and 8 are not available. It is also noted that the AMSU-A channel 3 for Metop-A/B/C has a higher NEDT value than the specification, which indicates that certain systematic performance issues remain with this channel.

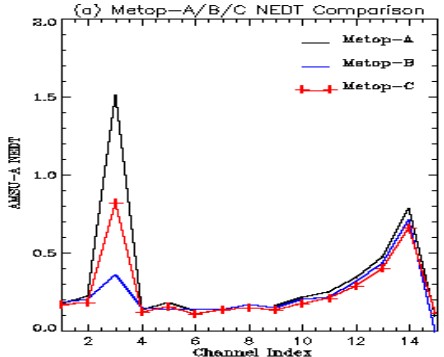 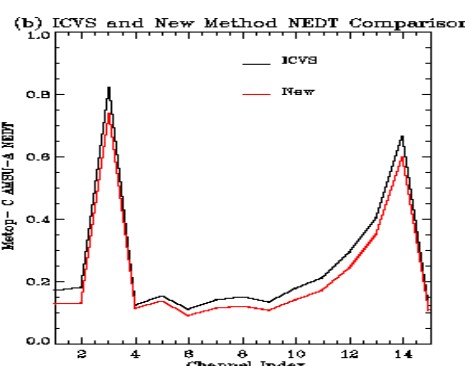

**Figure 6.** (**a**) NEDTs comparison on 15 November 2019 among Metop-A to -C AMSU-A instruments, where the NEDTs are computed using the Integrated Calibration/Validation System (ICVS) method and Metop-A AMSU-A channels 7 and 8 are not available. (**b**) NEDT value comparison estimated using the ICVS method and the new method.

Thirdly, the NEDT values estimated using the ICVS method are slightly higher than those using the new method (see Figure 6b) because of an overrated temperature sensitivity to warm counts [19]. Among all channels, the ICVS method produces relatively large errors in noise estimation in the first three channels compared with the new method. For example, the ICVS method causes an absolute error of 0.07 K in channel 3. The upper temperature sounding channels 10–14 are important for applications in NWP models especially. The ICVS method overestimated an error of around 0.05 K in those channels. In other words, the new method improves the accuracy of the noise estimate by 0.05 K. More discussions on the new method are conducted in [19].

Overall, channels 1–2 and 4–15 have demonstrated a stable noise performance within the specification since the launch. However, channel 3 displays an unstable noise feature and its NEDT constantly failed to meet the specification due to highly fluctuating warm counts and degraded channel gain over time.

## 5. AMSU-A TDR and SDR Quality Assessment

The quality assessment of Earth scene antenna (TDR) and brightness temperature (SDR) data are conducted by, respectively, using CRTM simulations and the inter-sensor comparison with legacy AMSU-A instruments flown on Metop-A and -B.

### 5.1. Comparisons with CRTM Simulations

This study focuses on a long-term stability assessment of the Metop-C AMSU-A TDR and SDR data quality by monitoring a one-year time series of AMSU-A observation (O) minus RTM simulation (B) differences from 15 November 2018 to 15 November 2019. Our observations represent either antenna temperatures ($T_A$) or SDR ($T_B$). The model simulations are computed using version 2.3 of the JCSDA CRTM [23,41,42], where we used the Fast Microwave Water Emissivity Model version 6 (FASTEM6) as the the oceanic microwave emissivity model [43,44]. The CRTM instrument characteristics for Metop-C AMSU-A are based on the specifications shown in Table 1 above. It is noted that the measured central frequency stability at channel 6 is from −4 to +10 MHz (not listed in Table 1) [27], slightly exceeding the upper limit of the specification (+5 MHz). The 10-MHz shift corresponds to the instrument temperature

at 263 K, which is lower than the on-orbit Metop-C AMSU-A1-1 instrument temperature (typically above 282 K). In addition, our sensitivity test also shows that the shift of 10 MHz causes an error in the order of 0.05 K when simulating brightness temperatures (the figure is omitted). Thus, the shift beyond the specification is neglected in the following simulations. As ancillary data of atmospheric and surface properties for the CRTM model, this study uses ECMWF analysis data for surface conditions and atmospheric profiles [45,46]. For consistency, the simulations were only carried out over oceans under clear skies for both window and sounding channels. A legacy algorithm for cloud liquid water content (LWC) estimates over oceans [47] is employed to exclude cloud-contaminated data, where LWC smaller than 0.1 mm is considered a clear sky condition.

For demonstration, Figure 7a–d display four types of results about Metop-C AMSU-A antenna temperature ($T_A$) and brightness temperature ($T_B$) biases against CRTM simulations for the data spanning from 15 November 2018 to 15 November 2019. The graph in Figure 7a is the yearly mean $T_A$ (black color) and $T_B$ (pink color) biases vs. the channel. Generally, the $T_A$ mean biases at sounding channels 4–14 are within −1 K, where the CRTM simulations are relatively accurate since they are less affected by errors in surface emissivity. However, the biases at three window channels (1, 2 and 15) and dirty sounding channel 3 are higher than 1.5 K. This inconsistency in the upper sounding channels is mostly due to RTM simulation errors because the simulation accuracy is very sensitive to errors in surface emissivity. For example, an emissivity error of 0.01 could cause an error in the order of 2 K at the abovementioned window and dirty sounding channels.

Compared with the $T_A$ biases, the $T_B$ biases are typically smaller for all AMSU-A channels except for the above window and dirty sounding channels due to inaccurate CRTM simulations. The reduced bias feature demonstrates the good performance of the conversion coefficients from TDR to SDR data. On the other hand, the standard deviations of all daily $T_A$ and $T_B$ mean biases during the same period are also included in Figure 7a, distributed from 0.05 to 0.3 K depending on the channel, with the largest standard deviation at channel 3. The relatively small standard deviation implies the decent stability of the data quality with time, while the largest standard deviation occurs at channel 3 due to its highly variable NEDT value with time. Regarding the standard deviation of the biases for all available pixels per day, they are large and are within the range from 0.2 K (upper sounding channels) to 2 K (window channels) (the figure is omitted).

The graph in Figure 7b illustrates the scan angle dependency of the yearly mean $T_A$ and $T_B$ biases at window channel 3 and sounding channels 5 and 10. It is well known that satellite microwave radiance (either $T_A$ or $T_B$) can show a strong angle-dependent feature towards the two ends of the scanning swath, partly due to changes in the optical path length through the Earth's atmosphere between the Earth and the satellite [48]. A certain angle dependency still remains within both $T_A$ and $T_B$ biases at all channels. As shown in (b), the $T_A$ biases from the nadir to the (left or right) end scanning positions show differences of more than 0.8 K for the sounding channels and more than 2 K for the window channel. The $T_B$ biases typically exhibit a reduced and more uniform scan dependent bias compared to $T_A$. For example, at channel 5, the $T_A$ biases change from −0.1 K at the nadir to −1.3 K at the right ending position (scan index 30), but the $T_B$ biases change from 0.2 K to −0.6 K. A similar angle dependency feature exists at other channels (the figure is omitted).

To give a full picture of the magnitude and angle dependency of the biases with time, Figure 7c,d display the time series of daily mean $T_A$ biases vs. time (X-axis) and scan position (Y-axis) for channels 3 and 5, respectively, covering the period from 15 November 2018 to 15 November 2019. Both Figure 7e,f are the same as Figure 7c,d except for the daily mean $T_B$ biases. Channel 5 has a stable bias pattern for both $T_A$ and $T_B$ along with angles and time, although the channel 3 bias is slightly variable with day, which is partially caused by the NEDT feature in Figure 5a. Again, the $T_B$ biases typically exhibit a reduced and more uniform scan-dependent bias compared to $T_A$, albeit the RTM simulation uncertainties remain at window channels. Similar conclusions are made at other channels (the figures are omitted). Currently, the derived antenna pattern correction (APC) coefficients have been delivered to a series of important users, including, but not limited to, the NOAA Microwave

Integrated Retrieval System (MiRS) [7], the NOAA Unique Combined Atmospheric Processing System (NUCAPS), the NOAA Environmental Modeling Center (EMC), the European Organisation for the Exploitation of Meteorological Satellites (EUMETSAT), the US Naval Research Laboratory (NRL), ECMWF, and the ATOVS (Advanced Television and infrared operational satellite Operational Vertical Sounder) and AVHRR (Advanced Very High Resolution Radiometer) Pre-Processing Package (AAPP).

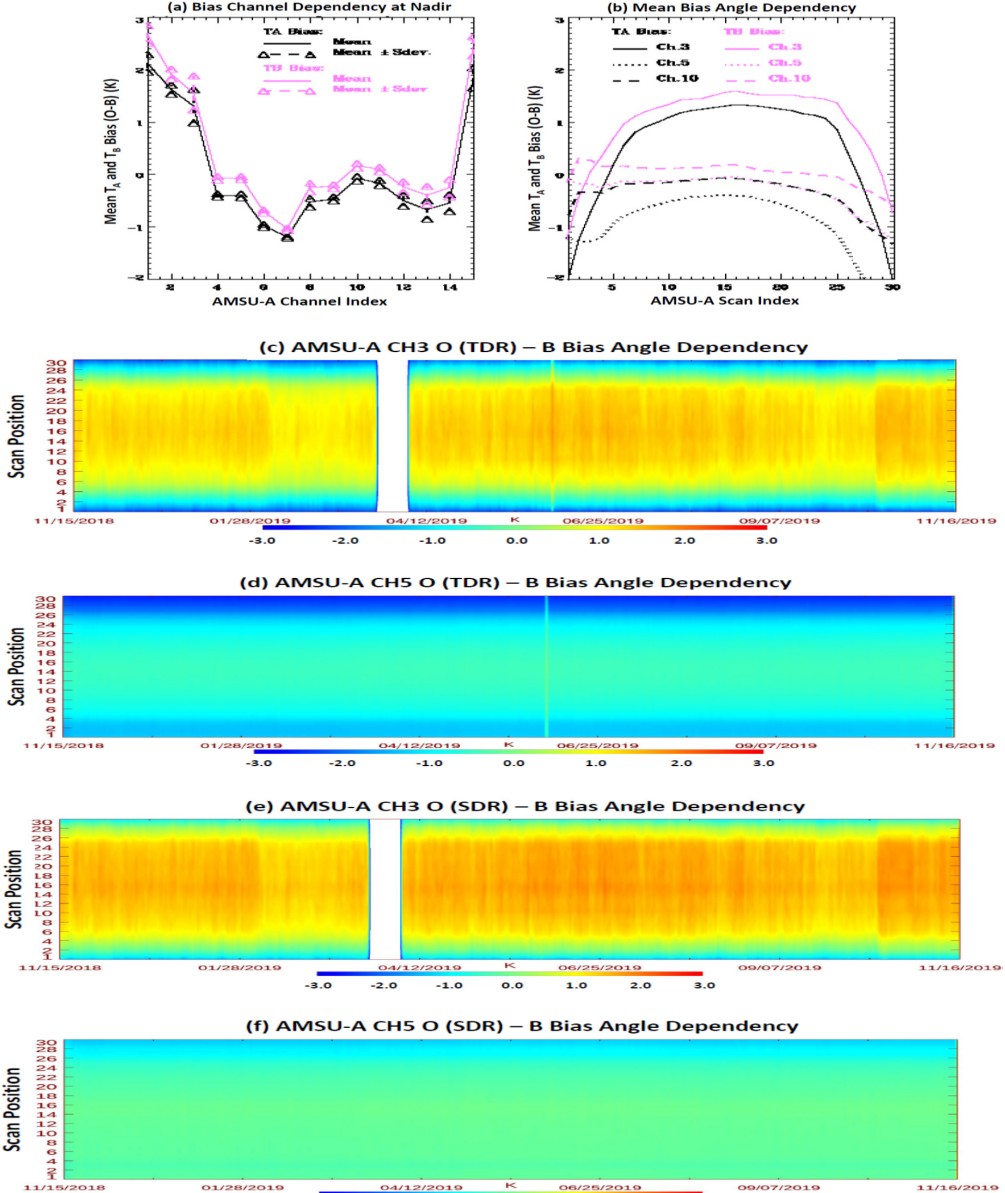

**Figure 7.** Metop-C AMSU-A antenna ($T_A$) and brightness ($T_B$) temperature biases against Community Radiative Transfer Model (CRTM) simulations from 15 November 2018 to 15 November 2019. (**a**) Yearly mean $T_A$ (black) and $T_B$ (pink) biases and standard deviation of all daily mean biases at the nadir direction vs. AMSU-A channel. (**b**) Yearly mean $T_A$ (black) and $T_B$ (pink) biases at channels 3, 5, and 10 vs. scan position. (**c**) Time series of channel 3 daily mean $T_A$ bias vs. scan position from 15 November 2018 to 15 November 2019. (**d**) Same as (**c**) except for channel 5. (**e**) Same as (**c**) except for $T_B$ bias. (**d**) Same as (**e**) except for channel 5.

The long-term stability of Metop-C AMSU-A TDR and SDR data quality has been validated by comparing the data to CRTM simulations from 15 November 2018 to 15 November 2019, showing a stable angular dependency feature against model simulations. Next, we investigated whether

Metop-C AMSU-A data quality is comparable to the data quality of legacy AMSU-A instruments flown on Metop-A and -B.

## 5.2. Metop-A, -B and -C Inter-Sensor Comparisons Using SNO Method

More than a decade ago, a technique was developed for accurately predicting the Simultaneous Nadir Overpasses (SNOs) of two Earth-orbiting satellites [24], which is referred as the SNO method. At each SNO, radiometers from both satellites view the same place at the same time at nadir, providing an ideal scenario for the intercalibration of radiometers aboard the two satellites. This technique was further improved to achieve the collocation of two passive-microwave satellite instrument SNO datasets with quality-controlled bilinear interpolation for window and surface-sensitive channels [25]. In this study, the inter-sensor comparisons among Metop-A, -B, and -C AMSU-A observations are performed based on double differences (DD) of SNO pairs between Metop-A, -B, and -C and each of NOAA-18 and -19 AMSU-A, where NOAA-18 or NOAA-19 AMSU-A is used as a transfer, as described below.

$$DD_{M_3-M_x}(N_{18}) = (M_3 - N_{18})_{SNO} - (M_x - N_{18})_{SNO}, \quad with \; x = 1,2 \tag{12}$$

and

$$DD_{M_3-M_x}(N_{19}) = (M_3 - N_{19})_{SNO} - (M_x - N_{19})_{SNO}, \quad with \; x = 1,2, \tag{13}$$

where $M_1$, $M_2$, and $M_3$ denote the Metop-B, -A, and -C individually for simplifying the length of the equations; and $N_{18}$ and $N_{19}$ are for NOAA-18 and -19, respectively.

All collocated AMSU-A SNO data sets are produced from the TDR data from Metop-A to -C and NOAA-18 and -19 from 30 November 2018 to 15 November 2019, all of which existed primarily in polar regions near 80° N and 80° S. To obtain more observations, each SNO pair is generated using 80-s temporal and 30-km spatial windows between two sensor observations. As discovered in previous studies, the large antenna temperature bias estimation uncertainties might remain within the SNO data sets for window and lower sounding channels, particularly over highly variable Earth scenes or cloudy conditions. Hence, an additional quality control (QC) is applied to check the inhomogeneity within field-of-view (FOV) for SNO pairs, as done in [26]. All pairs within an SNO event are removed from the collocated data sets if their standard deviation is greater than 2 K. Note that channels 7 and 8 for Metop-A AMSU-A and channel 15 for Metop-B AMSU-A are not operational during the selected data sets.

Figure 8a displays the averaged inter-sensor differences at 13 AMSU-A channels between Metop-C and Metop-A using either NOAA-18 (named N18 in the graphs for clarity) or NOAA-19 (named in the graphs as N19 for clarity) AMSU-A as a transfer. The results demonstrate that antenna temperatures from Metop-C AMSU-A are very comparable with those from Metop-A AMSU-A at the available channels. The differences (absolute values) at all channels, except for channel 3, are typically smaller than 0.3 K, by using either NOAA-18 or NOAA-19 AMSU-A as a transfer. Meanwhile, the differences are very comparable with two SNO references of NOAA-18 and NOAA-19 AMSU-A, except for channel 3. This is partially due to the large NEDT of Metop-A AMSU-A channel 3, which has a much high NEDT value (about 1.5 K), exceeding the specification and showing an unstable measurement performance.

Figure 8b shows the averaged inter-sensor differences at the 14 channels between Metop-C and Metop-B using either NOAA-18 or NOAA-19 AMSU-A as a transfer. Similar to the conclusion for Metop-C and -A, antenna temperatures from Metop-C AMSU-A are very comparable with those from Metop-B AMSU-A at all channels except for channel 15, which failed. The absolute differences at all available channels are typically smaller than 0.3 K and the differences are very comparable from two SNO references of NOAA-18 and NOAA-19 AMSU-A, except for channels 3 and 8. This deviation between two transfers is related to the noisy channel 8 of NOAA-19 with its high NEDT (0.9–1.2 K).

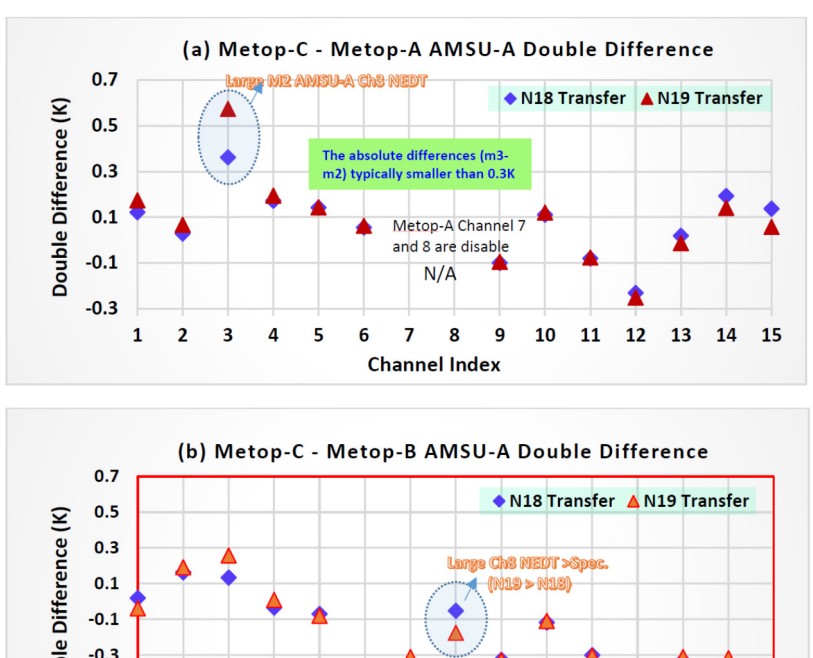

**Figure 8.** (**a**) Averaged inter-sensor differences at 13 AMSU-A channels between Metop-C and Metop-A using either NOAA-18 (named as N18 in the graphs for clarity) or NOAA-19 (named as N19 in the graphs for clarity) AMSU-A as a transfer. (**b**) Averaged inter-sensor differences at 14 AMSU-A channels between Metop-C and Metop-B using either NOAA-18 or NOAA-19 AMSU-A as a transfer.

Overall, the observed TDR and SDR data at all AMSU-A channels have shown a relatively stable quality since the launch. The higher NEDT at channel 3 has not had a critical impact on TDR and SDR data. The derived APC coefficients from TDR to SDR data have demonstrated a good performance in both deriving brightness temperatures and improving the asymmetrical bias features at most of the channels against the CRTM simulations. Moreover, the Metop-C AMSU-A data quality is comparable to Metop-A/B AMSU-A data, showing a decent quality and consistency with Metop-A/B AMSU-A.

## 6. Summary and Conclusions

This study presents an end-to-end Metop-C AMSU-A calibration and validation analysis. The calibration consists of the radiometric equation from Earth scene radiometric counts to antenna temperature and the conversion equation from antenna temperature to brightness temperature by removing side lobe contaminations resulting from cold space and satellite spacecraft. In the radiometric equation, the cold space temperature calibration correction due to antenna side lobe contaminations are derived using prelaunch antenna pattern functions, albeit the nonlinearity parameter is derived using the prelaunch TVAC data. Moreover, the optimal cold space view (SV) positions for Metop-C AMSU-A are determined based on initial OV data from 18 November 2018 to 30 November 2018, where SV1 (i.e., the satellite zenith angle of 83.3°) is determined for AMSU-A1 and SV3 (i.e., 80.0°) for AMSU-A2.

Next, the instrument noise performance is characterized using the NEDT, which is calculated primarily by using the current ICVS method [18] to enable a consistent analysis with that of legacy AMSU-A instruments, albeit the EUMETSAT and United Kingdom Met Office gain-based methods and a new method [19] are implemented for comparison. Channels 1–2 and 4–15 have demonstrated a stable noise performance within the specifications since the launch and up to 15 November 2019. However, channel 3 displays an unstable noise feature and is frequently higher than the specification

(recently in the order of 1.0 K) due to highly fluctuating warm counts and degraded channel gain with time. Regarding the accuracy of the NEDT estimation using the ICVS method, the ICVS method is found to overestimate the NEDT against the new method by approximately 1–10%, depending on the channel.

Finally, the quality of Metop-C AMSU-A TDR and SDR data is comprehensively assessed by using the CRTM simulations and inter-sensor comparison with legacy AMSU-A onboard Metop-A and -B. Against the CRTM simulations, Metop-C AMSU-A TDR and SDR data at all AMSU-A channels have shown a relatively stable quality since the launch. The higher NEDT at channel 3 has not caused a vital impact on TDR and SDR data quality. The derived APC coefficients have demonstrated a good performance in both deriving brightness temperatures and improving the asymmetrical bias features at most of the channels against the CRTM simulations. On the other hand, the inter-sensor comparisons between Metop instruments, via either NOAA-18 or NOAA-19 AMSU-A as a transfer, have demonstrated that Metop-C AMSU-A data quality is comparable to Metop-A/B AMSU-A data, showing that Metop-C AMSU-A fits into the family of Metop series AMSU-A instruments.

However, residual biases remain in the calibration process. Particularly, brightness temperature biases at some channels are not close to zero and show certain residual symmetric angle dependence, where the biases towards the two ends of the scanning swath are slightly different. This feature is a common issue for all AMSU-A instruments. A few radiation perturbation components could contribute to the residual biases, which are neglected in the TDR to SDR conversion algorithm, e.g., antenna emissions and heterogeneity effects due to the difference in the Earth's radiation at different viewing angles [5]. In addition, possible instrument polarization misalignment might be an additional cause of the asymmetric feature [49]. In addition, the current calibration equation (see Equation (1) or (2)) is established to derive the Earth scene radiance or antenna temperature by using the warm load temperature as the starting point in the interpolation. This approach becomes questionable if the warm load PRT temperature is unstable with time. For example, a couple of Kelvin variations have remained in Metop-C AMSU-A since the launch. This instability with time could result in some errors in the derived Earth scene antenna and brightness temperatures. Alternatively, the calibration equation should be revised to use the cold space temperature as the starting point in the interpolation. This is a common issue for all AMSU-A instruments. Therefore, it is worth conducting a separate study to understand these common issues in more depth and to further improve the AMSU-A TDR and SDR data quality.

**Author Contributions:** Conceptualization, B.Y., J.C., and C.-Z.Z.; methodology, B.Y., J.C., C.-Z.Z., and K.A.; software, B.Y., J.C., K.A., H.Q., and T.Z.; validation, B.Y., J.C., C.-Z.Z., and K.A.; investigation, B.Y., J.C., C.-Z.Z., K.A. and K.G.; data curation, D.H. and J.G.; writing—original draft preparation, B.Y.; writing—review and editing, B.Y., C.-Z.Z., K.G., and J.G.; visualization, B.Y., J.C.; supervision, B.Y.; project administration, B.Y.; funding acquisition, B.Y. and J.G. All authors have read and agreed to the published version of the manuscript.

**Funding:** This research was funded by the NOAA Office of Projects, Planning and Analysis (OPPA).

**Acknowledgments:** The authors would like to thank Walter Asplund and Michael Honaker of the NASA Goddard Space Flight Center for providing AMSU-A System In-Orbit Verification (SIOV) information for the optimal cold space position analysis. The useful comments of Changyong Cao are also gratefully acknowledged. Thanks also go to Ninghai Sun and Zhaohui Zhang for their contributions to the previous and current AMSU-A CRTM simulations. We would also like to thank the four anonymous reviewers for providing many valuable suggestions. Finally, yet importantly, thanks go to Northrop Grumman Electronic Systems for providing a number of technical reports, which are available upon request from the authors.

**Disclaimer:** The manuscript contents are solely the opinions of the author(s) and do not constitute a statement of policy, decision, or position on behalf of NOAA or the U.S. Government.

**Conflicts of Interest:** The authors declare no conflict of interest.

## Appendix A. Radiometric Calibration Counts (Blackbody and Cold Counts)

There are two samples of cold and warm count measurements per scan for AMSU-A1 and -A2 [27]. For each scan, the blackbody counts $C_W$ and the space counts $C_C$ are the averages of two samples of the internal black body and the space view, respectively.

$$\overline{C}_X(i) = \frac{C_{X_1}(i) + C_{X_2}(i)}{2}, \tag{A1}$$

where $C_X(i)$ (where $X = W$ or $C$) for the ith scan line. If any two samples differ by more than a preset limit of blackbody count variation $\Delta C_X$ (the initial limit is set to $3\sigma$, where the standard deviation, $\sigma$, is calculated from the prelaunch calibration data $C_X$ for each channel), the data in the scan should not be used. To further reduce the noise in the calibrations, $C_X$ (where $X = W$ or $C$) for each scan line is convoluted over several neighboring scan lines according to the weighting function [29]

$$\overline{C}_X = \frac{\sum_{i=-n}^{n} W_i C_X(t_i)}{\sum_{i=-n}^{n} W_i} \tag{A2}$$

where $t_i$ (when $i \ldots 0$) represents the time of the scan lines just before or after the current scan line and $t_0$ is the time of the current scan line. One can write $t_i = t_0 + i\Delta t$, where $\Delta t = 8$ s for AMSU-A. The $2n + 1$ values are equally distributed about the scan line to be calibrated. Following the NOAA-KLM operational preprocessor software, the value of $n = 3$ is chosen for all AMSU-A antenna systems. A set of triangular weights of 1, 2, 3, 4, 3, 2, and 1 are chosen for the weight factor $W_i$ that appears in Equation (A2) for the seven scans with $i = -3, -2, -1, 0, 1, 2$, and 3.

## Appendix B. Blackbody Target Temperatures

Radiances for both AMSU-A1 and -A2 Earth views are derived from the radiometric counts and the calibration coefficients inferred from the internal blackbody and space view data. The physical temperatures of the internal blackbody targets are measured by platinum resistance thermometers (PRTs). As shown in [29], the PRTs were calibrated against 'standard' ones traceable to the National Institute of Standards and Technology (NIST) to measure the temperatures of the internal blackbody targets and have an accuracy of 0.1 K. The outputs of the telemetry are PRT counts, which must be converted to PRT temperatures. The normal approach for deriving the PRT temperatures from counts is a two-step process, in which the resistance of each PRT (in ohms) is computed by a count-to-resistance look-up table provided by its manufacturer. Then, the individual PRT temperature (in degrees) is obtained from an analytic PRT equation. Here, this has been compressed to a single step in a polynomial form, with negligible errors, using an existing method [29], i.e.,

$$T_{Wk} = \sum_{j=0}^{3} f_{kj} C_k^j \tag{A3}$$

where $T_{Wk}$ and $C_k^j$ represent the temperature and count of each PRT. The coefficients $f_{kj}$ are provided for each PRT.

The mean blackbody temperature used in the calibration in Equation (1) (in the main body of the manuscript) $T_W$ is a weighted average of all samples of the PRT temperatures per scan:

$$T_W = \frac{\sum_{k=1}^{m} W_k T_{Wk}}{\sum_{k=1}^{m} W_k} + \Delta T_W \tag{A4}$$

where $m$ represents the number of PRTs for each antenna system and the scan index 'i' is omitted in the equation for clarification. For AMSU-A1, which includes channels 3–15, there are five measurement samples of warm load PRT temperatures per scan. For AMSU-A2, there are seven samples of

warm load PRT temperatures per scan [27]. $W_k$ is the weight assigned to each PRT and $\Delta T_w$ is the warm load correction factor for each channel, derived from the TVAC calibration data for three instrument temperatures (low, nominal, and high). Values for $\Delta T_W$ are provided for each instrument. For AMSU-A1-1, $\Delta T_W$ values for Phase Locked-Loop Oscillators (PLLO) #1 and PLLO #2 are provided separately. The $W_k$ value, which equals 1(0) if the PRT is determined to be good (bad) before launch, will be provided for each flight model. If any of the PRT temperatures $T_{Wk}$ differ by more than 0.2 K from their value in the previous scan line, then $T_{Wk}$ should be omitted from the average in Equation (A4).

## Appendix C. On-Orbit AMSU-A NEDT Methods

In the following descriptions, for clarity, the calculation method for AMSU-A instrument on-orbit NEDT in the ICVS is expressed as $NE\Delta T^{ICVS}$, whereas the new method in [19] is called $NE\Delta T^{New}$. A brief introduction without the channel index is given below, but detailed descriptions can be found in [18,19], correspondingly.

According to [18],

$$NE\Delta T^{ICVS} = \sqrt{\frac{1}{4(N-2)} \sum_{i=1}^{N-1} \frac{1}{\overline{G(i)}^2} \left[ \left( C_{W_1}(i+1) - C_{W_1}(i) \right)^2 + \left( C_{W_2}(i+1) - C_{W_2}(i) \right)^2 \right]} \qquad (A5)$$

with

$$\overline{G(i)} = \left| \frac{\left( \overline{C_W(i)} - \overline{C_C(i)} \right)}{\left( \overline{T_w(i)} - \overline{T_C(i)} \right)} \right|, \qquad (A6)$$

where $N$ is the number of scans per orbit; '$i$' is the scan index per orbit; $\overline{C_C(i)}$ *and* $\overline{C_W(i)}$ are the averages of two samples of cold and warm counts per scan, respectively, as defined in (A1); $\overline{T_W(i)}$ is the average of five samples (for AMSU-A1 channels) or seven samples (for AMSU-A2) of warm load PRT temperatures per scan; and $\overline{G(i)}$ is the averaged calibration gain per scan.

According to [19], the new NEDT method is described as follows.

$$NE\Delta T^{New} = \sqrt{\left( NE\Delta T_{C_W} \right)^2 + \left( NE\Delta T_{C_C} \right)^2 + \delta_{Cov(C_W,\,C_C)}} \qquad (A7)$$

where

$$\left( NE\Delta T_{C_W} \right)^2 = \frac{1}{4(N-2)} \sum_{i=1}^{N-1} \overline{\left( \frac{\partial T_A(i)}{\partial C_W(i)} \right)}^2 \left[ \left( C_{W_1}(i+1) - C_{W_1}(i) \right)^2 + \left( C_{W_2}(i+1) - C_{W_2}(i) \right)^2 \right], \qquad (A8)$$

$$\left( NE\Delta T_{C_C} \right)^2 = \frac{1}{4(N-2)} \sum_{i=1}^{N-1} \overline{\left( \frac{\partial T_A(i)}{\partial C_C(i)} \right)}^2 \left[ \left( C_{C_1}(i+1) - C_{C_1}(i) \right)^2 + \left( C_{C_2}(i+1) - C_{C_2}(i) \right)^2 \right], \qquad (A9)$$

$$\delta_{Cov(C_W,C_C)} = \frac{1}{4(N-2)} \sum_{i=1}^{N-1} \overline{\frac{\partial T_A(i)}{\partial C_W(i)}} \cdot \overline{\frac{\partial T_A(i)}{\partial C_C(i)}} \times \left\{ \sum_{k=1}^{2} \left( C_{W_k}(i+1) - C_{W_k}(i) \right) \left( C_{C_k}(i+1) - C_{C_k}(i) \right) \right\} \qquad (A10)$$

$$\overline{\frac{\partial T_A(i)}{\partial C_W(i)}} = \frac{\left( \overline{T_w(i)} - \overline{T_C(i)} \right) \left( \overline{C_C(i)} - \overline{C_S(i)} \right)}{\left( \overline{C_W(i)} - \overline{C_C(i)} \right)^2}, \qquad (A11)$$

$$\overline{\frac{\partial T_A(i)}{\partial C_C(i)}} = \frac{\left( \overline{T_w(i)} - \overline{T_C(i)} \right) \left( \overline{C_S(i)} - \overline{C_W(i)} \right)}{\left( \overline{C_W(i)} - \overline{C_C(i)} \right)^2}, \qquad (A12)$$

where $\overline{\frac{\partial T_A}{\partial C_W(i)}}$ *and* $\overline{\frac{\partial T_A}{\partial C_C(i)}}$ denote the scan-averaged derivatives. More detail about the above equations can be found in [19].

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
