# Peer review of "Calibration and Validation of Antenna and Brightness Temperatures from Metop-C Advanced Microwave Sounding Unit-A (AMSU-A)"

_remotesensing, doi:10.3390/rs12182978_

Round 1
Reviewer 1 Report
In this study, the authors carried out the calibration and validation of Antenna Temperature Data Record (TDR) and Brightness temperature Sensor Data Record (SDR) data from the last NOAA Advanced Microwave Sounding Unit-A (AMSU-A) flown on the Metop-C satellite. Overall, the paper is appropriate for Remote Sensing and I would recommend publication with addressing a few minor issues described in the following section.
1 Specific comments
-The introduction can be expanded with by adding sufficient background information and including more relevant references.
-The description of the AMSU-A calibration approaches is quite technical. It would be beneficial to readers to further describe the general ideas/concepts on which rely those methods.
-Please consider to improve the image quality of most figures (dpi, font size).
-Line 281: The terms “gradually” and “significantly” in one sentence sound a bit ambiguous. Please revise it in a better way.
-Line 438: . . . has not caused
Author Response
dear Reviewer,
Thank you for your valuable comments. Please see the attachment for our responses.
Banghua Yan

Reviewer 2 Report
The paper is very interesting and presents all very well the details of the AMSU instrument on MetOp-C.
I have just a few comments:
Line 127: What is the further analysis mentioned to obtain the data in table 2 from the data plotted in figure 2f
Figure 2 editorial: The subfigure c) and d) are included twice.
Figure 2: I was expecting the lunar contamination to be the spike visible in subfigure a) and b), while if I understand correctly, the spike is an outlier filter. Please confirm that in this case the lunar contamination doesn't change the plot between a) and b)
Section 5.2. it is not clear which data where used from MetOp-A and B for the comparison. The result is very interesting, however, it should be of interest to mention which data of MetOp-A and B were used, I mean: If data obtained at the same period in time as MetOp-C, which means with a certain aging for the other two or if you use historical data like during the SIOV or anyway at Beginning of life of MetOp-B and MEtOp-A.
The two appendix are really appreciated.
Author Response
Dear Reviewer,
Thank you for your valuable comments. Please see teh attachment for the responses from us.
Sincerely,
Banghua Yan

Reviewer 3 Report
This paper provides a good summary of the authors work on calibration and validation of the ASMU-A instrument on Metop-C. Much of this work is based on well established methods that have been developed for previous instruments. I do have
a few comments and questions on the work done where I believe the paper is incomplete for understanding how the methods
were applied here. I have also included several editorial comments.
1) Section 5.1: Uncertainty in the surface emissivity is cited as the reason for larger differences between CRTM simulations and the AMSU-A TDRs / SDRs. It would be helpful to include information regarding the version of CRTM used and the surface emissivity model and version used for the simulations.
2) Section 5.2: The paper should provide a more complete description of the SNO method used. The methods used in refs [9-11] differ. Ref [11] doesn't seem appropriate here since that paper addresses calibration of conically scanning instruments which have no nadir observations.
3) There is not enough information provided in the paper to understand the advantages of the new method for computing NEDT and ref [26] is not available. Please provide more information on this method.
4) There are a number of NOAA and Northrop Grumman technical reports listed in the paper such as [1-4], [8], [18] and [21]. These are often difficult to find. Please provide a URL for each of these if possible.
5) Table 1: The heading for column six reads "Measured 3-db Bandwidth". I believe this should be "Measured 3-dB Beamwidth". This also applies to the Table 1 footnotes. Also, formatting of the Table 1 headers should be improved.
6) Figure 2: Panels (c) and (d) are repeated in the paper. I do not see a difference between panels (a) and (b) -- is there now lunar contamination in this example or is it too small to see?
7) PLLO is defined in an appendix but should instead be defined in section 3.2 where it is first used.
8) line 251: "sessions" should be "sections"
9) line 460: The channel 3 is NEDT is higher than the spec but I don't think "superior" is the right word to use here.
10) The last paragraph of the conclusion is not clear to me. Is the "asymmetrical bias" referring to the scan angle dependent biases shown in Figure 7(b)? Also, isn't this issue common to all of the AMSU-A instruments? Please provide clarification in the text.
11) line 547: change "clarify" to "clarity"
Author Response
Dear Reviewer,
Thank you for your valuable comments. Please see the attachment for our responses. Please let us know if you have any new comments.
Thanks.
Sincerely,
Banghua Yan

Reviewer 4 Report
This paper is overall well written and important to the community. I only have a few minor comments,
- Figure 2, the label size is a little small. In addition, the Figure.2c and 2d were plotted twice.
- Line 173~183: RS should be Rs, in a similar manner, RSL -> Rsl, CS -> Cs, RW->Rw, RC->Rc. Please check all the symbols used.
- Misusing the subscripts of Rw, Rc, Cw, and Cc in equations 1-4? For example, “Rs = Rw + …” should be “Rs=Rc + …”. Please check these equations.
Author Response

(The authors gave the same response as above.)

Round 2
Reviewer 3 Report
The authors have adequately responded to my previous comments and revised the manuscript accordingly. I did note the following minor editorial issues:
line 66: "NPW" to "NWP"
line 67: sentence is awkward (delete "would"?)
line 295: missing the work "Phased"
Figure 5: Caption references four subplots but five are shown and the fourth seems to be the same as the third.
line 481: "SCO" to "SNO"
Author Response
Dear the reviewer,
Thank you for your valuable comments. Please see the attached for the responses to your comments.
Sincerely,
Banghua Yan
